# Building a pipeline to identify and engineer constitutive and repressible promoters

Eric J.Y. Yang and Jennifer L. Nemhauser

Department of Biology, University of Washington, Seattle, WA, USA

## Original Research Article

*Arabidopsis thaliana*; Boolean logic; constitutive promoter; *Nicotiana benthamiana*; synthetic biology.

**Corresponding author:**
Jennifer L. Nemhauser;
Email: jn7@uw.edu

## Abstract

To support the increasingly complex circuits needed for plant synthetic biology applications, additional constitutive promoters are essential. Reusing promoter parts can lead to difficulty in cloning, increased heterogeneity between transformants, transgene silencing and trait instability. We have developed a pipeline to identify genes that have stable expression across a wide range of *Arabidopsis* tissues at different developmental stages and have identified a number of promoters that are well expressed in both transient (*Nicotiana benthamiana*) and stable (*Arabidopsis*) transformation assays. We have also introduced two genome-orthogonal gRNA target sites in a subset of the screened promoters, converting them into NOR logic gates. The work here establishes a pipeline to screen for additional constitutive promoters and can form the basis of constructing more complex information processing circuits in the future.

## 1. Introduction

Plant synthetic biology aims to provide greater control over plant form and function, a goal that is beginning to be realised. Several projects have produced measurable gains in photosynthetic efficiency (Batista-Silva et al., 2020; Orr et al., 2017), and others have intervened in hormone response pathways to change plant architecture (Khakhar et al., 2018) or environmental response (Lim et al., 2020; Park et al., 2015). These advances rely on well-characterised promoters to ensure the expression of transgene in desired tissues.

Promoters can be broadly broken down into three categories based on expression pattern: constitutive, spatiotemporally restricted, and inducible (Peremarti et al., 2010). Constitutive promoters are defined here as promoters expressed in all tissues at all times. These promoters regulate the transcription of what are commonly referred to as 'housekeeping genes'. While each category of promoter is useful in plant engineering, constitutive promoters are often used to confer novel traits such as herbicide tolerance, to drive synthetic circuits, and used in metabolic engineering projects due to their broad tissue coverage (Bak & Emerson, 2020; Brophy et al., 2022; South et al., 2019). Some of the most widely used plant constitutive promoters include variants from the *Cauliflower Mosaic Virus* 35S (35S) promoter, and promoters from members of the ubiquitin and actin families (Jiang et al., 2018; Peremarti et al., 2010). However, the list of available plant constitutive promoters is short, and this lack of parts poses many challenges (Peremarti et al., 2010). Having to reuse the limited number of promoters in increasingly complex plant gene circuits or metabolic engineering projects can quickly lead to instability of the transformed construct due to repeated elements rearranging and homology-dependent gene silencing, which is heritable (De Wilde et al., 2000; Peremarti et al., 2010; Rajeev Kumar et al., 2015).

To expand the number of promoters available, several groups have recently used distinct strategies to engineer both constitutively and conditionally expressed promoters. One approach builds synthetic promoters by adding *cis*-elements to a 'minimal promoter region', which is often 35S-derived. By varying the number and type of *cis*-elements, researchers were able to generate promoters with a wide range of expression levels and expression patterns (Ali & Kim, 2019; Belcher et al., 2020; Brophy et al., 2022; Liu & Stewart, 2016). Another approach uses sequences upstream of the minimal promoter region as a landing dock for synthetic activators guided by zinc-finger, TALE, or dCas9 to promote expression (Liu & Stewart, 2016). The expression strength of these promoters can be tuned by varying the number of target sites for

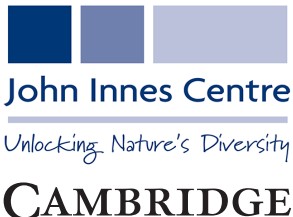

the synthetic activators (Cai et al., 2020; Moreno-Giménez et al., 2022). These approaches, while quite powerful, are limited by the small number of characterised minimal promoters available to build upon and may still lead to repeated units in large constructs if the same minimal promoters were used.

Here, we employed an alternative approach for finding constitutive promoters. Instead of building and testing synthetic promoters, we looked to natural promoters found in the *Arabidopsis* genome. This approach has a few advantages. Synthetic promoters require extensive characterisation to determine their expression pattern and, because of practical constraints, are often only tested in a few selected tissues. In contrast, the wealth of RNA-seq data available for *Arabidopsis* provides highly detailed information about a given promoter's likely expression potential, including the expression level of the gene throughout many developmental stages, tissue types, and even various growth and/or stress conditions. The expression of a native promoter has already been subject to selective pressures, and so is potentially more likely to remain stable across generations. By introducing a set of unique sequences, natural promoters also have the potential to minimise the likelihood of gene silencing or unwanted recombination through repeated units in multigenic constructs. Lastly, by employing some of the techniques in generating synthetic promoters described above, these native promoters could potentially form the foundation for suites of derived promoters with even more refined expression levels. A similar approach successfully expanded the range of promoter expressions available in *B. subtilis* (Guiziou et al., 2016). Since the argument for the need of additional promoter parts can be directly extended to the need for additional terminators, and terminators are known regulators of gene expression (Andreou et al., 2021; Wang et al., 2020), we screened promoter-terminator pairs together. To further extend the utility of the new promoter/terminator pairs, we also introduced dCas9 target sites with sequences not found elsewhere in the *Arabidopsis* genome, thereby enabling specific repression by synthetic transcription factors without interfering with the cognate native genes.

## 2. Results

To identify the most stably expressing promoters available in the *Arabidopsis* genome, we analysed publicly available RNAseq datasets. The majority of the RNAseq dataset came from the Klepikova transcriptome profile which included multiple tissues from different development stages (Klepikova et al., 2016). We supplemented this dataset with an RNAseq dataset for pollen (Loraine et al., 2013), as this cell type was not represented in the Klepikova dataset. After processing the RNAseq datasets, there were 10,096 genes that were expressed in all the datasets (i.e. have none zero read counts) (Figure 1a). The coefficient of variation (CV) of expression across different tissues is often used as a metric for identifying stably expressed genes (Czechowski et al., 2005; Huang et al., 2019; Wang et al., 2019). Within the lowest 3% CV, there were 303 genes, which corresponds to a CV cutoff of 0.26 (Figure 1a,d). To facilitate dissemination of the parts quantified in this study, we adopted the Golden Gate MoClo system and cloned the promoter + 5′UTR together as a standard MoClo part and similarly with 3′UTR + terminator (Figure 1c; Engler et al., 2014; Weber et al., 2011). Since MoClo uses BsaI and BbsI type-II restriction enzymes for cloning, we removed any candidates with these restriction sites within the cloned regions. These cloning constraints left us with 61 candidate genes.

To selectively activate or repress promoters in the context of a synthetic circuit, we wanted to modify segments of the promoter region to allow genome-orthogonal dCas9 targeting. While there are no specific guidelines on optimal placement for gRNA target sites in plants (Pan et al., 2021), studies in other eukaryotes have pointed to −50 to +300 bp from TSS in mammalian cells for CRISPRi, and within −200 bp from TSS in yeast (Jensen, 2018). Using the 'Binding Site Prediction' function from PlantRegMap we screened for predicted motifs within 500 bp of the promoter region from the TSS (Tian et al., 2020). We retained promoters that could accommodate two 23 bp gRNA target sites (20 bp target sequence and 3 bp PAM site) without interrupting any predicted motifs and were at least 67 bp apart, following the spacing used in Gander et al. (Figure 1b). We were left with 33 candidate genes. Compared to the commonly used native *Arabidopsis* constitutive promoters, the candidates identified here were more stably expressed but have mostly weaker mean expression (Figure 1d). Detailed information of the 33 candidates can be found in Supplementary Table S2.

While one of the main goals of this study is to identify the best available natural stable genes through analysis of RNAseq data, the 'stability' of the candidate genes we screened for in this article is constrained by the choice of RNAseq dataset used. The Klepikova dataset included stress-treated leaf samples with heat, cold, and wounding treatments, but they were not included in the CV calculation since the samples were only collected from mature third leaves and no other tissue types. Instead, we normalised the stress data with untreated 'mature whole third leaf' and calculated their CV and included the result in Supplementary Table S2 for reference. Similarly, while the datasets capture coarse temporal resolutions throughout development, they cannot identify the fluctuation of circadian genes and therefore we supplemented the final table with identified circadian genes from CGDB for reference (Li et al., 2017).

Of the 33 stably expressed genes identified from the bioinformatics pipeline, we successfully cloned 22 promoter-terminator pairs. We tested the promoters in *Nicotiana benthamiana* (tobacco) transient agroinfiltration assays and identified 16 promoters that had expressions that were significantly different from the negative control (Figure 2).

To determine whether the promoters showed constitutive expression in *Arabidopsis*, 12 of the promoters were selected to drive expression of the RUBY reporter (He et al., 2020) in stable transformants. Since RUBY is a pigment that allows for simple visual readout, we were hoping it would be an effective way of evaluating the expression of the promoters in all the tissues throughout development. Three representative T1 lines were selected for each construct and six T2s per T1 line were observed at the seedling stage (12 days) and as mature plants (day 34). Eleven of the 12 promoters transformed showed expression in *N. benthamiana*, yet we only identified three promoters that displayed RUBY expression in *Arabidopsis*. Representative individuals are shown in Figure 3a (Supplementary Figure S3) with the intensity of RUBY colouring quantified in Fiji (Supplementary Figure S4).

Given that expression in *N. benthamiana* doesnot perfectly predict expression in *Arabidopsis*, we included two promoters (AT1G54080, AT1G71860) that did not show expression in tobacco infiltration in our *Arabidopsis* stable transformation experiment. Interestingly, AT1G54080 displayed RUBY expression in roots and pollen. AT5G37830 had visible expression in pollen, siliques, stems, and roots. AT3G08530 had the most ubiquitous expression and had visible expression in the flowers, pollen, siliques, stems, and roots. A visual summary of the *Arabidopsis* and *N. benthamiana* experiments can be found in Supplementary Figure S5.

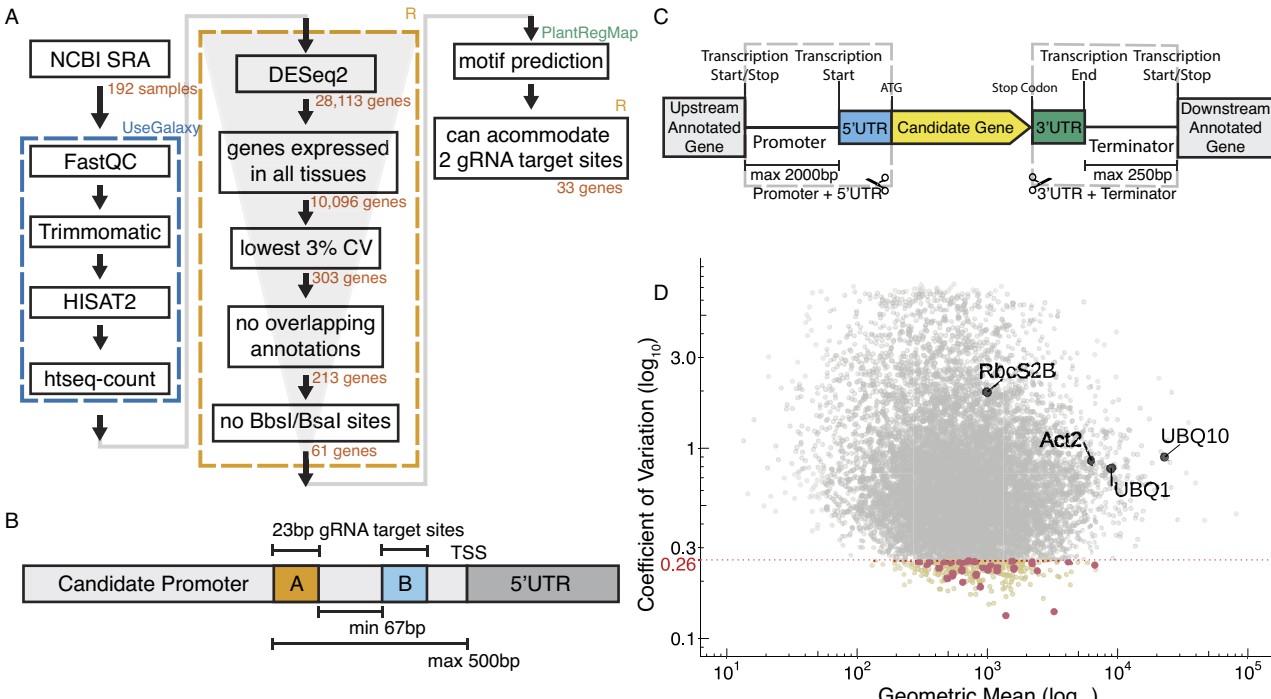

**Figure 1.** (a) Pipeline to identify constitutive promoters. The number of genes that pass each filter is indicated, along with the software used to implement the analysis. SRA is the 'Sequence Read Archive'. Detailed methods, including parameters for each filter, are described in Section 4. (b) Schematic of filters used to select candidate promoters to engineer with synthetic gRNA target sites. (c) Schematic describing how we defined 'promoter' and 'terminator'. The 'promoter' was defined here as starting from the transcription start site and going upstream to a maximum of 2,000 bp or to the next annotated neighbouring gene, whichever is shorter. Similarly, a 'terminator' was defined as starting from the transcription end site and going downstream a maximum of 250 bp or to the next annotated neighbouring gene, whichever is shorter. Promoters and terminators were cloned, along with their respective UTRs, following the Golden Gate MoClo system. (d) Plot showing values for the 10,096 genes expressed in all tissues. The geometric mean of expression across samples is plotted on the *x*-axis with the coefficient of variation (CV) on the *y*-axis. Both axes are on a base-10 log scale. The lowest 3% CV corresponds to a 0.26 CV cutoff, and the 303 genes with CV lower than 0.26 are highlighted in yellow. The final 33 candidates that fulfilled all criteria are highlighted in red. Several common promoters used in plant synthetic biology are annotated for reference.

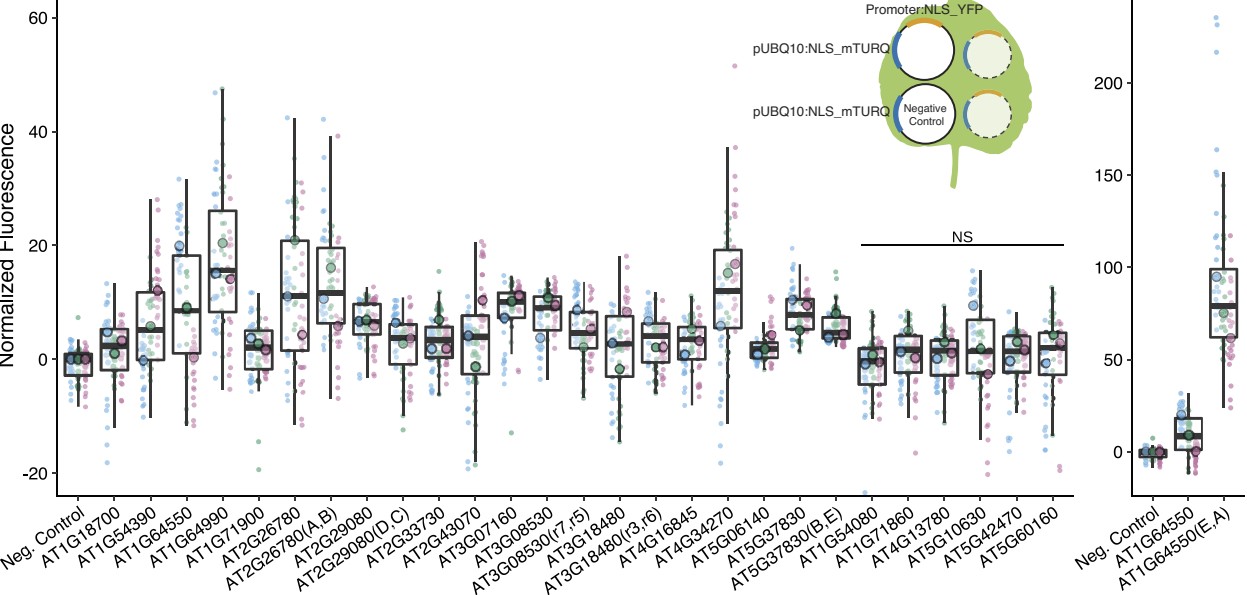

**Figure 2.** We identified 16 promoters that expressed in *N. benthamiana*. Six promoters were modified to introduce gRNA target sites. These sites are designated by brackets following the gene name. Three different constructs were injected per leaf, each containing a promoter to be tested driving NLS_YFP and an internal control of pUBQ10:NLS_mTURQ. Each leaf also has a negative control injection that only contains pUBQ10:NLS_mTURQ. Normalisation is performed using the formula:
$\frac{\mathrm{YFP_{promoter}} - \mathrm{median}\left(\mathrm{YFP_{Neg.control}}\right)}{\mathrm{mTURQ_{promoter}}}$ . For each construct, the three replicates with median fluorescence levels closest to the median of the group were selected for visualisation and statistical analysis. Each biological replicate is represented by a beeswarm plot of 24 datapoints (12 per leaf disc, 2 disc per injection) collected from the plate reader as well as a single summarising datapoint representing the median. The boxplots represent all biological replicates. Significance test was performed using Dunnett's test for comparing multiple treatments with control at 95% family wise confidence level. Non-significant constructs are marked as NS. For a given construct, the colours signify datapoints derived from the same biological replicate.

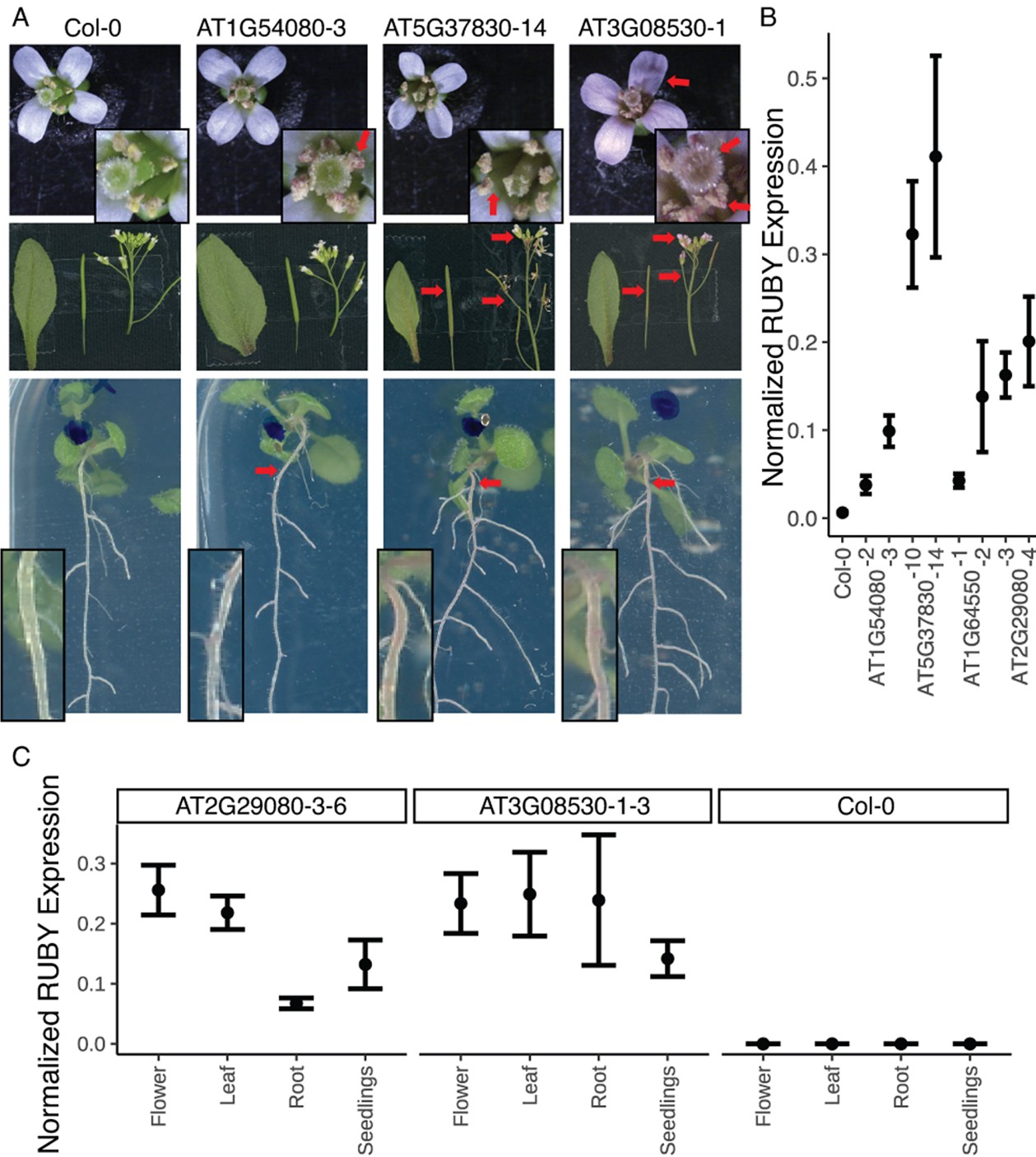

**Figure 3.** (a) Three promoters showed expression of RUBY in *Arabidopsis* T2 plants. The flowers, siliques, and leaves were imaged on day 34, while the seedling images were imaged on day 12. The inset boxes are the same images at higher magnification. Red arrows indicate areas where RUBY expression is visible by the eye. (b) qPCR data on T2 whole seedlings in three biological replicates for each line and (c) qPCR data on tissues collected from T3 plants, each with three biological replicates and two technical replicates. RUBY expression was normalised against the reference gene PP2AA3, and the bars represent the mean expression of the RUBY reporter and the standard error of the mean (SEM).

Given that the majority of the promoters had no visible expression of RUBY by eye, we performed qPCR on whole seedlings from four promoter lines: two with visible RUBY expression in roots (AT1G54080 and AT5G37830) and two without (AT1G64550 and AT2G29080). The two lines without visible RUBY expression both had qPCR expression level between the two lines with visible RUBY expression in the roots. This result suggests that RUBY was not a reliable reporter for these low-expressing promoters and that the promoters were indeed functional in *Arabidopsis* (Figure 3b).

To further confirm whether the promoters were truly constitutively expressed, we chose our strongest expressing line (Figure 3a) and one of the lines that did not appear red but had detectable expression by qPCR (Figure 3b) for more careful qPCR analysis on seedlings, adult roots, flowers, and leaves (Figure 3c). The result confirms that RUBY expression was detected in all the tissues analysed, even when the tissues might not appear red by visual inspection. An interesting observation from the qPCR experiments was that the expression level of RUBY mRNA detected through

qPCR is weaker than expected from the RNAseq dataset. While the predicted expression level of all four of the genes in the qPCR experiment are higher than the reference gene PP2AA3, the measured result showed the opposite (Supplementary Table S7). This discrepancy could be attributed to the RUBY reporter or potential limitations in identifying additional transcriptional regulators (see Section 3).

To make the promoters screened in this experiment more versatile, we next introduced two gRNA target sites into six of the promoters screened with target site sequences not found in the *Arabidopsis* genome. A constitutive promoter with two unique gRNA target sites can function as a NOR gate (a two-input logic gate where the output is only ON when neither of the inputs is present) in the presence of a dCas9-guided repressor. The inputs for such a gate are the gRNAs. When either or both of the gRNAs are present, the dCas9-guided repressor should be able to keep the promoter OFF (Figure 4b). Only when neither of the guides is present can the promoter be turned ON. Nine different functional gRNA sequences (A–I) were selected from the literature (Supplementary Table S8).

We first confirmed that the introduction of the target sites did not abolish promoter expression (Figure 2). While in most cases there was little difference in expression between modified and native promoters, in one case (AT1G64550(E,A)), the expression level increased dramatically, possibly due to the introduction of new TF binding sites at the junction of the introduced gRNA target-site (Supplementary Figure S6). The repressibility of the modified promoters was tested in *N. benthamiana* with each infiltration containing all constructs required for repression (Figure 4a). Each set of experiment contains four possible input combinations for each repressible promoter and the extent of repression was evaluated against the non-repressed control using two non-matching gRNAs (Figure 4c,d). Five of the modified promoters (mPromoters) functioned as NOR gates while AT3G18480(F,G) acted as a NOT gate with input2 (gRNA_G) (Figure 4d). Of the NOR gates, AT2G26780(A,B) repressed to similar extents with either or both inputs. AT1G64550(E,A), AT2G29080(D,C) and AT3G08530(H,I) all displayed additive effects where having both inputs gave stronger repression than just having one alone. AT5G37830(B,E) had a well-repressed target site with input2 (gRNA_E) while input1 (gRNA_B) alone resulted in a weaker repressed state. The strongest repression was observed for AT5G37830(B,E) and AT1G64550(E,A) with about a twofold repression between input(1,1) and input(0,0), while the rest of the promoters had around a 1.2-fold repression. The repression strength observed in the assay is quite modest, and it is likely because the promoters are quite weak to begin with, making strong repression more difficult. The result displayed as normalised fluorescence and not as fold repression can be found in Supplementary Figure S9.

## 3. Discussion

Constitutive promoters are essential staples in stocking the synthetic biology toolbox. They are versatile due to their wide expression coverage, and form the foundation from which many synthetic promoters are built. Here, we report on the establishment of a pipeline to find the most stably expressing promoters in *Arabidopsis*. We successfully used this approach to identify 16 promoters that are predicted to be more stably expressed than some of the most widely used native plant constitutive promoters, and showed they can drive expression in transient transformations of *N. benthamiana*. We attempted to capture the expression pattern of these promoters in stably transformed *Arabidopsis* using the visual RUBY

reporter and uncovered limitations in its utility, but we identified at least two promoters that showed expression in all the tissues tested throughout development via qPCR. Lastly, we engineered repressible versions of six promoters and showed that five of these can function as NOR logic gates.

One of the biggest challenges in having a small selection of promoters to choose from is the need to reuse promoters in larger constructs, which could pose challenges to long-term stability. The promoters identified in this article were selected from some of the most stably expressed genes available in the *Arabidopsis* genome and all have distinct sequences. A lack of promoter parts also means a lack of flexibility when it comes to the range of expression strength. Most of the promoters used in plant synthetic biology are quite strong and that is not ideal for every application. The availability of weaker, broadly expressed promoters like those characterised here allows more flexibility in promoter choices when excess production of target proteins can be a problem. For example, they can be beneficial in avoiding toxic intermediates or optimising flux in metabolic engineering projects (Brückner et al., 2015; Patron, 2020). If a minimal promoter sequence can be identified from these native promoters, they can also serve as the foundation of additional synthetic promoters where the expression pattern and strength can be freely modified by adding *cis*-elements or synthetic transcription factor binding sites. The pipeline employed in this article to arrive at new native constitutive promoters should be readily adaptable to other organisms if there is sufficiently broad sampling of transcriptomes and a reference genome. The pipeline could also be modified to identify native promoters with particular expression patterns. One caveat is that the promoters that can be extracted in this way are, by definition, limited by what is naturally available in the organism. On the other hand, they have the advantage of already being assayed in a whole range of tissue types and developmental stages – a breadth of information that can be logistically challenging to collect for synthetic promoters. It will be interesting to see if synthetic devices made with these modified native promoters prove more resilient to mutation than those using fully engineered promoters, as these sequences have presumably maintained stable expression in the face of mutation and selection.

Evaluating the promoters using RUBY revealed that the novel reporter had limited sensitivity when driven by weaker promoters. We were able to detect RUBY expression in seedlings and adult tissues without visible colouration using qPCR, a more sensitive assay. However, it is important to note that detecting transcripts doesnot always imply comparable levels of protein production due to post-transcriptional and post-translational regulation. In our design, we attempted to capture the effects of any post-transcriptional regulation by including the UTRs, but other potential transcriptional regulators could be missed. The lower-than-expected RUBY mRNA levels detected could be due to such regulators. Promoter-proximal introns after the translation start codon, for example, would not be captured in the cloning pipeline though it is known to contribute to gene expression (Rose, 2019; Rose et al., 2008). Distally located regulatory regions would also not be captured, but they should be rare in the compact genome of *Arabidopsis* (Galli et al., 2020; Lu et al., 2019).

Working with native promoters also provided an opportunity to learn more about the biology of promoters themselves. Yamamoto and colleagues suggested that plant promoters can be grouped into a few core promoter categories based on the presence or absence of certain location-sensitive motifs (Yamamoto et al., 2011). Interestingly, they reported that TATA-box containing promoters tend

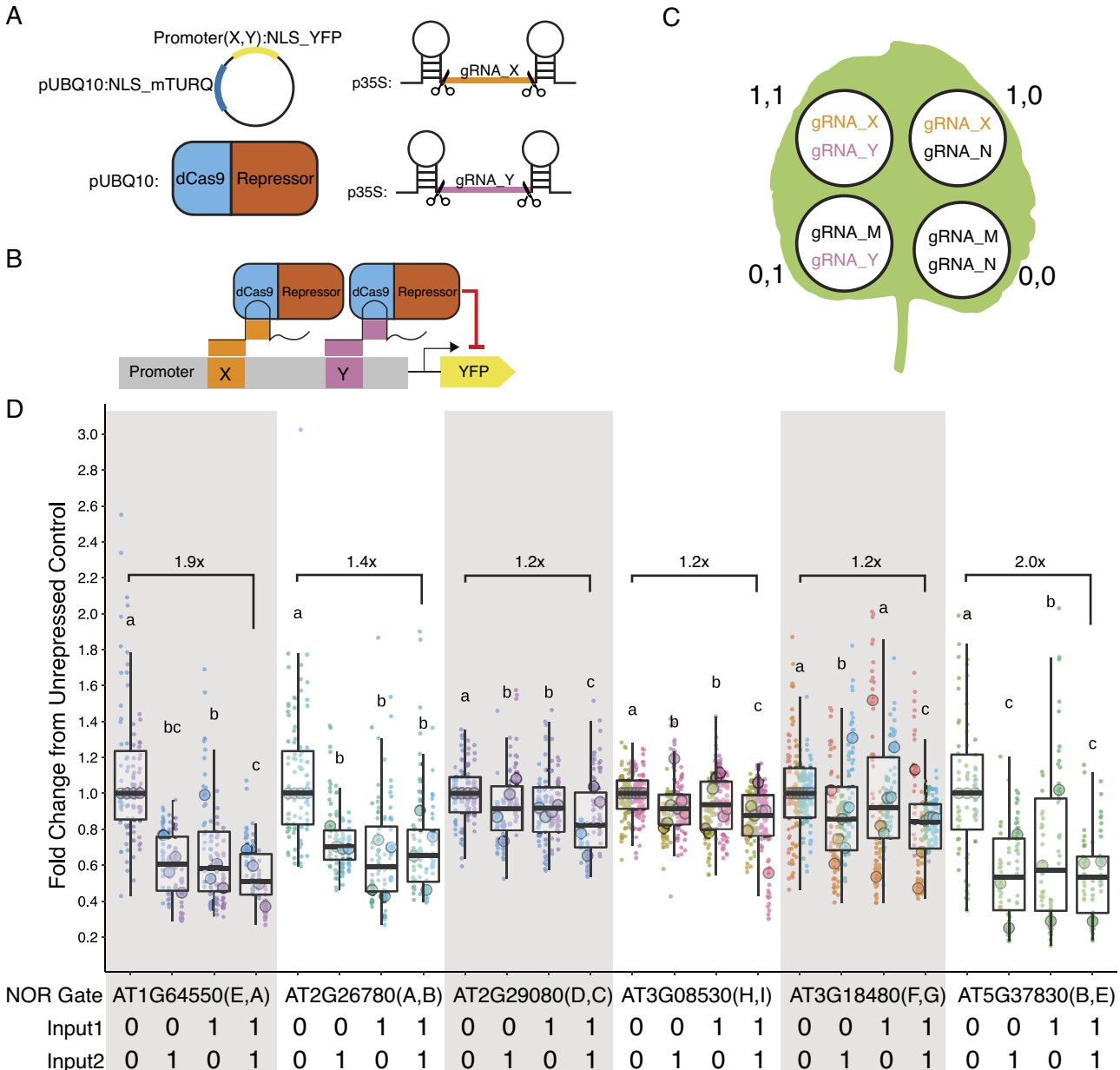

**Figure 4.** (a) The four constructs co-injected for each injection. The injection always contains the mPromoter, dCas9-guided repressor, and the two self-cleaving input gRNAs. The gRNAs are denoted with *X* and *Y* representing a variable input. (b) Schematic of the NOR gate when both input gRNAs are present. (c) Pattern of injection for the four possible input combinations and the gRNAs used for each injection. (1,1) represents both guides are present while (0,0) represents neither is present. When a guide is not present, a non-matching gRNA is injected in its place, denoted here as gRNA_M or N. (d) Five of the six mPromoters functioned as NOR gates. All guides apart from gRNA_F are independently repressible. Each biological replicate is represented by a beeswarm plot of 24 datapoints (12 per leaf disc, 2 disc per injection) collected from the plate reader as well as a single summarising datapoint representing the median. The boxplots represent all biological replicates. The signal is measured as the YFP fluorescence (driven by the promoters being tested) divided by the mTURQ fluorescence (driven by pUBQ10). In each set of NOR gate injections, the (0,0) injection serves as the unrepressed control, and the dataset is normalised by dividing all values by the median of the unrepressed control on a per-leaf basis. The *y*-axis represents fold changes from the unrepressed control and each biological replicate of the control is centred on 1. Each colour represents a unique leaf. Letters above each boxplot are Compact Letter Display (CLD) for all pairwise comparisons within each set of injections using ANOVA followed by Tukey's Honest Significant Difference Test. Numbers above the boxplot represent fold repression between the (0,0) and (1,1) injections.

to be regulated promoters while Coreless promoters (promoters that do not have any characteristic location-sensitive motifs) tend to be constitutively expressed. The vast majority of the constitutive promoters used today in plant synthetic biology are from the TATA promoter class, and we also have a much better understanding of how their expression is regulated (Cai et al., 2020). If the goal is to find constitutive promoters, however, the analysis by Yamamoto and colleagues would suggest that we should look to Coreless promoters instead. Indeed, only 9% (3/33) of the candidate genes

identified in this study contain TATA boxes, while 45% (15/33) are Coreless (Supplementary Table S2).

The ability to selectively activate and repress genes provides the tools necessary to perform Boolean logic, which would allow more complex computations (Kassaw et al., 2018). Plants naturally perform complex computations to determine when and where a gene should be expressed by integrating internal and external signals, and genetic logic gates provide a modular way to synthetically construct these input–output relationships by using simple

genetic parts. There are many ways to achieve the different logic operations using molecular biology (Patron, 2020). A NOR gate is powerful in that it can be used to construct any logic gate by just stringing together multiple NOR gates, and its efficacy has been demonstrated in yeast (Gander et al., 2017). To date, the feasibility of building more complex logic circuits in plants has been hindered by the lack of unique and strongly repressible promoter parts. With just our design constraints and no additional refinement, five of the six NOR gates built showed the correct behaviour, suggesting the pipeline used holds promise in identifying additional promoter candidates for engineering. The repressible promoters evaluated in this work can be further improved through additional design-build-test cycles to optimise the individual gRNA target sites by varying their position and sequence. The repressor design can also be potentially improved upon to lower the overall OFF state. While *N. benthamiana* serves as a great prototyping platform, the performance of the gates would also need to be evaluated in stable *Arabidopsis* lines to validate their viability. The pipeline and the repressible promoter screened in this work contributes to the construction of more complex, synthetic plant logic operations in the future.

## 4. Methods

### 4.1. Downloading and processing RNA-seq datasets

We used a custom UseGalaxy pipeline to process the RNA-seq datasets (Afgan et al., 2016). SRR accession codes from BioProject IDs PRJNA314076 (138 samples; Klepikova et al., 2016), PRJNA268115 (20 samples; Klepikova et al., 2015), PRJNA324514 (32 samples), PRJNA194429 (2 samples; Loraine et al., 2013) were input into 'Faster Download and Extract Reads in FASTQ (Galaxy Version 2.10.8+galaxy0)' with default settings. The FASTQ files were pipped into 'FastQC (Galaxy Version 0.73+galaxy0)' and 'Trimmomatic (Galaxy Version 0.38.0)' with sliding window trimming averaging across 4 bases with required average quality 20, and a minimum read length of 36. The trimmed files were input into 'HISAT2 (Galaxy Version 2.1.0+galaxy5)' with reference genome assembly TAIR10 and Araport11 genome annotation from The *Arabidopsis* Information Resource (TAIR). Minimum intron length was set to 60, and maximum intron length was set to 6000 (Marquez et al., 2012). Features from the Araport11 annotation were counted with 'htseq-count (Galaxy Version 0.9.1)' set to Union Mode and counting only reads within regions defined as 'exons' in the Araport11 annotation while not counting non-unique/ambiguous reads (Klepikova et al., 2016). The counted features were downloaded, and subsequent analysis was done in R (R Core Team, 2022).

### 4.2. Identifying stable promoters

All samples excluding the stress dataset (PRJNA324514) were normalised using the Median Ratios method from the DESeq2 package in R (Love et al., 2014). The coefficient of variation (CV) for each gene was calculated from the normalised data. Genes with the lowest 3% CV were kept for further analysis. The stress dataset from PRJNA324514 was normalised with 'mature whole third leaf' from PRJNA314076 for CV calculation, separate from the rest of the data.

### 4.3. Extracting promoter and terminator sequences

Promoter+5′UTR region (from before the start codon and extending upstream till the first annotated neighbouring gene or to a maximum of 2 kb from the transcription start site, whichever

is shorter) and 3′UTR+terminators (from after the stop codon and extending downstream till the first annotated neighbouring gene or to a maximum of 250 bp past the transcription end site, whichever is shorter) of the remaining genes were extracted using the Araport11 genome annotation and the '3,000 bp upstream and downstream' sequence files from the TAIR website. The extracted sequences were screened for BbsI and BsaI restriction enzyme cut sites and only those without were kept. Any genes with their promoter+5′UTR and 3′UTR+terminator overlapping annotations from neighbouring genes in the Araport11 annotation were also removed.

### 4.4. Transcription factor binding site prediction

The promoter sequences of the remaining genes were uploaded onto PlantRegMap using the 'Binding Site Prediction' function (Tian et al., 2020) and the predicted motifs for each promoter sequence were downloaded. Only genes that can fit two 23 bp gRNA target sites at least 67 bp apart without interrupting any of the predicted motifs while being within 500 bp of the TSS were kept.

### 4.5. Annotating candidate genes

For the final 33 candidate genes, CV for the Stress Dataset (StressCV) and promoter and terminator sequences were extracted as described above. The promoter core type was annotated by Tokizawa et al. (2017). A list of experimentally determined circadian genes in *Arabidopsis* was downloaded from CGDB (Li et al., 2017), and any UniprotKB identifiers were converted to ATG identifiers with the Uniprot Retrieve/ID mapping tool. Gene Descriptions (Representative Gene Model Name, Gene Description, Gene Model Type, Primary Gene Symbol, and All Gene Symbols) were retrieved from TAIR.

### 4.6. Construction of plasmids

Promoter+5′UTR and 3′UTR+terminator for each candidate genes as defined above were cloned with PCR from extracted genomic Col-0 DNA into their respective MoClo level zero acceptors (pICH41295 and pICH41276, respectively) (Engler et al., 2014). The promoter and terminator pair of the candidate genes was paired with nuclear-localised Venus to make level one constructs in 'position one'. Venus level one constructs were paired with pUBQ10 promoter driving nuclear localised mTURQ with an Act2 terminator from the MoClo Plant Parts Kit (pICH44300) in 'position two' to form ratio-metric lvl2s with a binary Ti vector backbone (pAGM4673 or pAGM4723). RUBY from (He et al., 2020) was cloned into level zero constructs and then cloned directly into level-2 Ti vector backbone with the promoter and terminator pairs (pICH86966). List of primers and plasmid maps can be found in Supplementary Tables S10 and S11 and Genbank files can be found in Supplementary Data S13. Plasmids used in this manuscripts are available at https://www.addgene.org/Jennifer_Nemhauser/with IDs 205359 - 205408.

### 4.7. gRNA target-site introduction

gRNA target sites were cloned into regions that do not disrupt any predicted TF binding sites (as described above) through Gibson assembly by replacing the original sequence (Gibson et al., 2009). Primers can be found in Supplementary Table S10.

### 4.8. Agrobacterium infiltration

In total, 5 mL cultures of *Agrobacterium* containing constructs to be injected along with a separate 25 mL culture of P19 (Win &

Kamoun, 2004) were grown overnight at 30C with the appropriate antibiotics. On the following day, the overnights were centrifuged at 3,000 × g for 10 minutes. The pellets were resuspended with 1 mL MMA (10 mM $MgCl_2$, 10 mM MES (pH 5.6), 100 uM acetosyringone). The OD of the cultures were measured and about 1~2 mL volume mixture with 5.0 OD for construct to be tested and 5.0 OD for P19 were prepared. The infiltration mix was rotated to mix at room temperature for 3 hours before injecting into fully emerged *N. benthamiana* leaves with a 1mL syringe. The injections were always injected as triplicates on three separate leaves on three separate tobacco plants. Each leaf is also always injected with a pUBQ10:mTURQ control.

## 4.9. Fluorescence quantification in N. benthamiana

At 3 days post infiltration, the leaves were clipped off and visualised in the Azure C600 Western Blot imaging system with exposure times Cy5 = 0 sec, Cy3 = 15 sec, Cy2 = 5 sec. Two hole punches were taken out of representative regions of each injection, and damaged regions with high background fluorescence were avoided. The leaf discs were placed in a 96-well plate on top of 200 uL of water, and the plates were read with a TECAN SPARK plate reader with YFP: excitation 506(15) and emission 541(15), Gain 100. mTurq: excitation 430(15) and emission 480(15), Gain 50. mScarlet: excitation 565(15) and emission 600(15), Gain 100. Settings: Multiple Reads Per Well; Circle (Filled) 4 × 4 with border 800 uM. Each leaf disc was read 12 times giving a total of 24 datapoints per injection per leaf. The output data was read into a custom R file for annotation, clean-up and visualisation. Multiple biological replicates were assayed for each promoter being tested, and the three replicates closest to the median of all replicates were kept for visualisation and statistical analysis. Each injection's YFP value is subtracted by the median YFP value of the pUBQ10:mTURQ negative control on the same leaf. The YFP value is then divided by the mTURQ value for each injection to normalise across results. A Dunnett Test, a post hoc pairwise multiple comparison test from the DescTools package, was used to determine whether the injections were significantly different from the negative control (Signorell et al., 2022).

## 4.10. Repression assays

Repression assays were performed using the modified promoters (mPromoters) with gRNA target-sites driving NLS-YFP and pUBQ10:NLS-mTURQ internal control as the reporter. The mPromoters (5OD) were co-injected with P19 (1OD), TPL repressor (1OD), self-cleaving gRNA_1 (1OD), and self-cleaving gRNA_2 (1OD). To test the mPromoters' NOR gate functionality, two gRNA inputs were required. In cases where only one input is present, the other self-cleaving gRNA will be a non-matching guide to the mPromoter. When neither inputs are present, sometimes two non-matching gRNAs (1OD each) were co-injected and sometimes only one (2OD), but the final total OD were always consistent. The exact injection combinations can be found in the R script. The TPL repressor construct contains pUBQ1:tdTomato-pUBQ10:dCas9_TPL(N188) and is modified from Khakhar et al. (2018). Self-cleaving gRNAs were designed in accordance with Zhang et al. (2017), and the modifications (gRNA and complementary sequences) were introduced in one step using Q5 mutagenesis (NEB) and were placed in the MoClo pICH86988 acceptor with a 35S promoter. The four possible input combinations for the NOR gate for each promoter were always injected on the same leaf, and the result was read with a plate reader as described above.

The YFP value of each injection was divided by the mTURQ value to normalise the data, and the value of each injection was divided by the median of the no-input control. An ANOVA followed by a Tukey's Honest Significant Difference Test was used to determine significant differences in expression between samples. List of plasmid maps used can be found in Supplementary Table S11 and Genbank files in Supplementary Data S13.

## 4.11. RUBY expression in Arabidopsis

Constructs with the candidate promoters driving RUBY were transformed into Col-0 through floral dipping method (Clough & Bent, 1998). T1 seeds were selected on 0.5× LS + 50 ug/mL Kanamycin+0.8% bactoagar. Plates were stratified for 2 days, light pulsed for 6 hours then kept in the dark for 3 days. Resistant seedlings were transplanted to soil to collect T2 seeds. For each promoter lines, three representative T1 lines were chosen to have their T2 seedlings phenotyped, and for each line, 19 T2 seeds were plated on 120 × 120 × 17 square petri dishes with 0.5× LS + 0.8% bactoagar without selection. The plates were imaged on day 4, 8, and 12 post-germination, and six representative seedlings were transplanted to soil. The plants were imaged with a digital camera on day 34. The flowers were imaged under a Leica S8AP0 dissecting scope. A representative leaf, a segment of the inflorescence, and silique were placed between two clear projector sheets and scanned with a flatbed scanner.

## 4.12. RUBY redness quantification

Images of were loaded into Fiji (Schindelin et al., 2012), and then converted to Lab stack to isolate the a∗ stack. The default colour of the extracted stack was green, so the image was further converted to an RGB stack so that the Green-channel could be used for region of interest (ROI) quantification.

## 4.13. qPCR

T2 seedlings were grown vertically on 0.5 × LS + 0.8% Phytoagar and without selection. The plates were stratified at 4°C for 2 days. On day 12, approximately five seedlings per replicate were frozen in liquid nitrogen. Three biological replicates were prepared for each genotype. Col-0 seedlings were also collected on day 12 as a single biological replicate. For T3 tissues, seedlings were grown on vertical 0.5 × LS + 0.8% Phytoagar plates without selection. On day 10, three sets of four seedlings were collected for each line and frozen in liquid nitrogen. Four seedlings per line were transplanted to fresh LS + Phytoagar plates to collect older roots from, and the rest of the seedlings were transplanted to soil. On day 22, three whole roots were collected from plants on plates and the tissues were frozen. The entire inflorescence and one leaf from three plants on soil were collected for each line between days 27 and 31. The tissues were collected in 2 mL tubes and powdered with a metal bead using a Retsch MM400 shaker after freezing the samples in liquid nitrogen. RNA was purified using an Illustra RNAspin Mini Kit (GE Healthcare). 1 $\mu$g of extracted RNA was then used with the iScript cDNA synthesis kit (BIO-RAD). qPCR was performed using the iQ SYBR Green Supermix (BIO-RAD). PP2AA3 was used as a reference gene and the primers for PP2AA3 and RUBY can be found in Supplementary Table S1. The standard curves were established using a pool of all the cDNAs. T2 RUBY seedlings were run with three biological replicates per line while Col-0 seedlings were run as four technical replicates. T3 tissues were run with three

biological replicates per tissue and two technical replicates per line. The qPCR was performed on a C1000 Thermal Cycler (BIO-RAD) and the result was read using the Bio-Rad CFX Maestro software and analysed using standard methods (Pfaffl, 2001).

## Acknowledgements

We thank Wesley George, Cassandra Maranas, Dr. Román Ramos Báez, Dr. Sarah Guiziou and Dr. Alexander Leydon for careful reading of the manuscript, as well as other members of the Nemhauser, Imaizumi and Steinbrenner labs for their feedback on this project. We thank Dr. Nicholas J. Provart for his help with the RNA-seq datasets.

**Funding statement.** This work was supported by the National Institute of Health (R01-GM107084), the National Science Foundation (IOS-1546873) and a Faculty Scholar Award from the Howard Hughes Medical Institute.

**Competing interest.** The authors declare no competing interests.

**Author contribution.** Experimental design and analysis: E.J.Y.Y., J.L.N. Research: E.J.Y.Y. Writing: E.J.Y.Y., J.L.N.

**Data availability statement.** The codes and datasets used in this study can be found on GitHub at https://github.com/Nemhauserlab/StablePromoters, and on Zenodo with DOI: 10.5281/zenodo.8170303. The repositories contain all the raw data as well as scripts to annotate, normalise and generate the figures used in the article. Datasets before annotation and annotated data before normalisation are both available. To minimise the supplemental file size, scripts without the datasets can be found in Supplementary Data S12.

**Supplementary material.** The supplementary material for this article can be found at https://doi.org/10.1017/qpb.2023.10.

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
