## [Reviewer Report]

13 October 2022

Dr. Olivier Hamant

Editor-in-Chief

Quantitative Plant Biology

Dear Dr. Hamant:

Attached please find our manuscript entitled “Expanding the synthetic biology toolbox with a library of constitutive and repressible promoters”, which we are submitting for consideration as an article. None of the material has been published or is under consideration elsewhere. We have submitted a preprint: MS ID#: BIORXIV/2022/511673.

For the ambitious engineering projects currently under development in plants, one of the major bottlenecks is the lack of well-characterized, constitutive promoters. Reusing promoter parts in large genetic constructs makes cloning challenging and increases the likelihood of transgene silencing, among other concerns. In our manuscript, we describe the development and implementation of a pipeline to identify broadly expressed genes. We then cloned out presumptive promoters and terminators of these loci, and validated their expression in both transient (Nicotiana benthamiana) and stable (Arabidopsis thaliana) transformation assays. To further increase the functionality for building complex circuits, we engineered a subset of these promoters with unique gRNA target sites and successfully generated orthogonally repressible NOR gates. The constitutive promoters screened in this study can help meet the need for additional promoter parts, and by converting them into NOR gates, they form the basis of constructing more complex logic gates in the future.

Based on the reception of this work by our colleagues in workshops and meetings, we believe it will appeal to a diverse range of scientists interested in transcriptional regulation, networks and synthetic biology. It would be fantastic if the debut of this toolset could be in an open-access journal like Quantitative Plant Biology.

Some potential reviewers for this work include:

Jenn Brophy, Stanford, jbrophy@stanford.edu, expertise: plant synthetic biology, esp. logic gates

Kevin Cox, Danforth Center, KCox@danforthcenter.org, expertise: transcriptional regulation

David Ehrhardt, Carnegie Institute, dehrhardt@carnegiescience.edu, expertise: vector and tool development

Naomi Nakayama, Imperial College London, n.nakayama@imperial.ac.uk, expertise: plant synthetic biology; transcriptional regulation, esp. promoters and terminators

Diego Orsáez, IBMCP, CSIC Valencia, dorzaez@ibmcp.upv.es, expertise: plant synthetic biology; dCas9-based tool development

Nicola Patron, Earlham Institute, nicola.patron@earlham.ac.uk, expertise: plant synthetic biology; transcriptional regulation, esp. promoters

Ross Sozzani, NCSU, rsozzan@ncsu.edu, expertise: vector and tool development; bioinformatics

Yoshiharu Y. Yamamoto, Gifu University, yyy@gifu-u.ac.jp, expertise: transcriptional regulation, esp. promoters

Sincerely,

Dr. Jennifer Nemhauser, jn7@uw.edu

Department of Biology

University of Washington

---

## [Reviewer Report]

Your manuscript has been now assessed by three reviewers, please apologize the delays. I share with them the opinion that your manuscript “Expanding the synthetic biology toolbox with a library of constitutive and repressible promoters“ is very interesting, timely and potentially relevant for the community. However, several valid concerns have been raised. Among other points, and in particular, the reviewers identified that the choice of RUBY as a reporter is not ideal, including the limitations shown in terms of sensitivity and that the dynamic range is relatively low. In addition, the experimental data doesn´t seem to support the claim of constitutive activation for the promoter. Further comparisons and in silico studies are suggested.

I would therefore be glad to consider publication of your article provided that it undergoes a revision tackling the issues raised by the reviewers

---

## [Reviewer Report]

21 May 2023

Dr. Olivier Hamant

Editor-in-Chief

Quantitative Plant Biology

Dear Dr. Hamant:

Attached please find our revised manuscript entitled “Building a pipeline to identify and engineer constitutive and repressible promoters”. We thank the editor and reviewers for their careful reading of our work. We have made a number of changes described below in response to the reviewer comments (including modifying the title), as well as adding new experimental results. We believe that these changes improve the manuscript and increase the accessibility of our findings. We hope that you agree.

Sincerely,

Dr. Jennifer Nemhauser

Department of Biology

University of Washington

jn7@uw.edu

Detailed response to reviews:

Reviewer 1 (R1)

Major Comments:

1) Title is too big to express the contents. Development of an Expression and Repression System for Transgenic Plants, for example, would be more appropriate.

Our response: Inspired by the reviewer’s suggestion, we changed the title to: Building a pipeline to identify and engineer constitutive and repressible promoters. We hope that this change captures the spirit of the suggested revision.

(R1) 2) Function of gene repression would be the key of the report. However, the demonstrated repression in Fig. 4D is much less effective than expected. This is not like repression but small modulation. The authors should try to develop ten fold repression or more. Otherwise, I don’t think of beneficial applications of the system.

Our response: While we understand (and share) the reviewer’s frustration with the relatively subtle impact of the repression we observe, it is worth noting that these parts have not been optimized (e.g., guide sites have not been moved around within the promoter), and that the overall low level of expression makes strong repression quite difficult to achieve. We have added text throughout the manuscript to emphasize the proof-of-concept nature of this study, and its shortcomings. Despite these issues, we strongly believe that these results will be of use to the plant synthetic biology community, as well as those interested in transcriptional regulation more generally.

(R1) Minor Comments:

1) Fig. 2 and 3. A positive control (e.g. 35S promoter - NOS terminator) should be included to see expression levels of the developed promoter-terminator pairs.

Our response: We can appreciate the reviewer’s desire to have a familiar promoter to serve as a point of comparison; however, we think that the levels of a 35S promoter are far too high to serve as a meaningful control for these endogenous, low-expressing promoters.

(R1) 2) Fig. 4D. Degree of repression should be shown. This is important information, and some researchers are really interested in it.

Our response: Thank you for this suggestion. We have modified Figure 4D accordingly.

(R2) Reviewer: 2

Major comments:

Although these promoters appear to have constitutive activity based on their expression patterns from transcriptomics data, it is not clear from the data shown in this paper that the 2 kb of upstream sequence (promoter) is capable of conferring constitutive activity in transgenic plants. For this reason, I suggest that the authors remove any statements that these promoters are constitutive, unless experiments are done in transgenic plants where a different (perhaps more sensitive) reporter gene is used (e.g., GUS) and gene expression is assessed in different tissues throughout the plant, at different developmental stages. Alternatively, RT-qPCR could be done on mRNA from individual tissues from the transgenic plants, at different developmental stages, to show constitutive activity.

Our response. We thank the reviewer for improving the precision of our language. We have now changed language throughout the text to differentiate between the objective of our screen versus what we have observed with our promoters. Additional qPCR had also been performed as per reviewer 2 and reviewer 3’s suggestion to confirm the expression of RUBY in the various tissues collected for two of the promoters. (Figure 3C had been added).

(R2) (Given the large variations in Tobacco data)... It is not clear that these NOR logic gates would function robustly in stable transgenic plants. I suggest this limitation be discussed.

Our response: We have added text to the Discussion to incorporate this suggestion as follows:

“While N. benthamiana serves as a great prototyping platform, the performance of the gates would also need to be evaluated in stable Arabidopsis lines to validate their viability.”

(R2) Minor suggestions:

(R2) Line 90: in the legend for Figure 1, define SRA as “Sequence Read Archive”.

Our response: We thank the reviewer for the suggestion, and have made this edit.

(R2) Lines 144-146: Twelve promoters were selected to transform Arabidopsis, yet one of those did not show activity in N. benthamiana? Why was this one promoter tested in Arabidopsis? This is not clear.

Our response: We have added a clarifying sentence regarding the promoter in question: “Given that expression in N. benthamiana doesn’t perfectly predict expression in Arabidopsis, we included two promoters (AT1G54080, AT1G71860) that did not show expression in tobacco infiltration in our Arabidopsis stable transformation experiment. Interestingly, AT1G54080 displayed RUBY expression in roots and pollen.”

(R2) Line 173: I could not find any instances of panels A and C from Figure 4 being mentioned in the text.

Our response: We thank the reviewer for the suggestion, and have added callouts for these panels.

(R2) Line 190: provide full gene number – AT1G64550(E,A)

Our response: We have made this correction.

(R2) Line 199: fluorescence

Our response: We have made this correction.

(R2) Line 217: As in my comments above, the reference to these promoters being constitutive should be removed.

Our response: We thank the reviewer for the suggestion, and have made this edit.

(R2) Lines 323-324: I don’t believe this sentence is referencing the correct Supplementary file numbers.

Our response: We have made this correction.

(R2) Line 329: …into regions that do not disrupt…

Our response: We have made this correction.

(R2) Line 345: Azure C600 Western Blot

Our response: We have made this correction.

(R2) Lines 387-388: Were these seeds plated on media without selection? T2 plants should still be segregating for the transgenes, so why were these grown without selection? How do you know that a plant not expressing RUBY is not because it doesn’t contain the transgene?

Our response: The seedlings were screened without selection to minimize stress on the seedlings. To ensure that at least some proportion of our seedlings did contain the transgene, we examined at least 19 T2 individuals per T1-line (independent insertion event).

Reviewer: 3 (R3)

Main considerations

(R3) The endogenous nature of the selected sequences prompts a question about their conservation, and thus transferability, to other species. The authors do address this when they test the promoters in transient tobacco agroinfiltration, however there is potential information being missed by such an approach. I wonder whether this could be addressed with an in silico approach, at least for the 5-6 promoters which are validated experimentally: for example can synteny of conserved transcription binding sites be compared across Arabidopsis accessions and other brassica species (for which the genome sequence is available). Similarly, the position where to insert gRNA targeted-sequences could be inspected for variability (with the assumption that high variability implies negligible regulatory capacity).

Our response: We thank the reviewer for this suggestion. We are also interested in exploring this cross-species promoter analysis. However, we believe the question is best addressed using a broader bioinformatic approach including many additional species, which would be a significant amount of additional work and is outside the scope of the current manuscript.

(R3) Ruby usefulness is questioned, based on the results showed in fig 3. However, the authors do not consider potential post-transcriptional regulation that the UTRs included in their design might entail. A comparison between qPCR and protein levels (if any antibody is available to test it) would be very informative.

Our response: We thank the reviewer for this suggestion and have added the aspect of post-transcriptional regulation in the discussion section:

“Evaluating the promoters using RUBY revealed that the novel reporter had limited sensitivity when driven by weaker promoters. We were able to detect RUBY expression in seedlings and adult tissues without visible coloration using qPCR, a more sensitive assay. However, it is important to note that detecting transcripts doesn’t always imply comparable levels of protein production due to post-transcriptional and post-translational regulation. In our design, we attempted to capture the effects of any post-transcriptional regulation by including the UTRs, but other potential transcriptional regulators could be missed. The lower-than-expected RUBY mRNA levels detected could be due to such regulators. Promoter-proximal introns after the translation start codon, for example, would not be captured in the cloning pipeline though it is known to contribute to gene expression (Rose, 2019; Rose et al., 2008). Distally located regulatory regions would also not be captured, but they should be rare in the compact genome of Arabidopsis (Galli et al., 2020; Lu et al., 2019).”

In regards to protein level analysis, we appreciate the suggestion though we are unaware of an antibody for RUBY.

(R3) Figure 2 shows that the selected promoters/terminators regulate genes with a range of expression levels. How does this compare between the endogenous genes and transgenes (analysed in figure 3)?

Our response: We thank the reviewer for suggesting this analysis. We have added a brief discussion of this as well as an additional supplementary table addressing the comparison between endogenous gene and transgene levels.

“The expression level of RUBY mRNA detected through qPCR is weaker than expected from the RNAseq dataset. While the predicted expression level of all four of the genes in the qPCR experiment are higher than the reference gene PP2AA3, the measured result showed the opposite (Supplementary Table S6). This discrepancy could be attributed to the RUBY reporter or potential limitations in identifying additional transcriptional regulators (see discussion).”

(R3) Finally, the constitutive nature of the promoters is only evaluated at the spatial level (and to a lesser extent, at the developmental one), but not their resilience to alteration by environmental stimuli (e.g. light/temperature). This could be tested in the T2 lines.

Our response: We thank the reviewer for their suggestion. Our main design specification was to identify broadly-expressed promoters, and our selection criteria do not include stress-resistance, therefore we have only evaluated these promoters on their expression pattern. We did, however, include a metric for stress-resistance in our Supplementary_Table_S2 that was determined from RNAseq data.

Minor considerations:

(R3) The second paragraph sets out the goal of this study and provides a definition for constitutive promoters. The term ubiquitous could be presented as well, as a tissue specific promoter could still be constitutively active at most/all developmental stages, but I appreciate that there is ambiguity in the literature about these terms.

Our response: We appreciate the reviewer noting the ambiguity in the use of the term “constitutive.” We have modified our text to state explicitly our definition: “Constitutive promoters are defined here as expressed in all tissues at all times”

(R3) The third paragraphs describes the previous approaches to generate constitutive promoters. By the authors own definition of ‘constitutive’, promoters with different expression patterns could not be defined as such. Similarly, the use of exogenous ZF, TALEN or Cas binding sites does not seem to address the problem, as their effectiveness would depend on the abundance of the trans-acting factor, moving the issue of regulation one level up.

Our response: We have modified the text to reflect this suggestion: “To expand the number of promoters available, several groups have recently used distinct strategies to engineer both constitutively and conditionally expressed promoters”

Results

(R3) Can the authors justify why limiting the analysis of transcription factor binding sites to 500 bp? Is regulation potential expected to decrease with distance from TSS?

Our response: We have added a few references to clarify this point: “While there are no specific guidelines on optimal placement for gRNA target-sites in plants (Pan et al., 2021), studies in other eukaryotes have pointed to -50 to +300bp from TSS in mammalian cells for CRISPRi, and within -200bp from TSS in yeast (Jensen, 2018)”

(R3) Figure 2 legend: from how many leaves does the beeswarm plot of datapoint collected from the platereader?

Our response: We have added text to the figure legend and methods section to clarify this point.

“For each construct, the three replicates with median fluorescence levels closest to the median of the group were selected for visualization and statistical analysis. Each biological replicate is represented by a beeswarm plot of 24 datapoints (12 per leaf disc, 2 disc per injection) collected from the plate reader as well as a single summarizing datapoint representing the median. The boxplots represent all biological replicates.”

(R3) Figure 3: the text (l155) mentions expression in the pollen but the resolution of the photo does not allow to see that (at least, not in the version available for review). More in general Ruby production is assessed locally, but it is unclear which tissue was used for RNA extraction before the qPCR analysis. I believe it would be interesting to repeat these qPCR in (at least some) different tissues, separately, and to compare transgene expression with the Ruby phenotype. It would be interesting to compare this information with transgenic lines where Ruby is driven by traditional constitutive promoters, if such transgenic plants are available.

Our response: We have made a clarification in the figure legends that the qPCR data was collected from T2 whole seedlings. Additional qPCR had also been performed as per the reviewer 2 and reviewer 3’s suggestion to confirm the expression of RUBY in the various tissues collected for two of the promoters. (Figure 3C had been added).

(R3) Figure 4. The legend states that colours represent datapoints from a single leaf (if I understand correctly). It seems to be a lot of datapoints from a single leaf, while in the analysis showed in figure 2 comes from leaves which were infiltrated in only 4 spots.

Our response: We have added text to the figure legend and methods section to clarify this point.

Discussion:

(R3) In the introduction, the authors correctly indicate the need to identify new minimal promoters, to design new full promoters that respond to specific stimuli. They could propose that such an analysis should be undertaken in the future on the promoters they identified in this study.

Our response: We thank the reviewer for their suggestion and have modified our discussion section accordingly

“If a minimal promoter sequence can be identified from these native promoters, they can also serve as the foundation of additional synthetic promoters where the expression pattern and strength can be freely modified by adding cis-elements or synthetic transcription factor binding sites.”

Materials and methods:

(R3) A description of how the gRNA target sequences where introduced in the cloned promoters is missing.

Our response: We have added text to the methods section to clarify this point. “gRNA target-sites were cloned into regions that does not disrupt any predicted TF binding sites through Gibson assembly by replacing the original sequence (Gibson et al., 2009)”

---

## [Reviewer Report]

Your revised manuscript has now been evaluated by the two of the three previously involved reviewers. I apologise for the delays. Thanks for your efforts preparing a new version which is indeed improved. You have tackled most of the scientific issues. I am pleased to accept the manuscript for publication provided that you tackle the minor comments of Rev#1 on Figs 2 and 3. I´m glad to be able to bring you encouraging news and I look forward to receiving the revised manuscript.

---

## [Reviewer Report]

23 July 2023

Dr. Olivier Hamant, Editor-in-Chief

Dr. Matias Zurbriggen, Associate Editor:

Quantitative Plant Biology

Dear Dr. Hamant and Dr. Zurbriggen:

Thank you for the wonderful news about the acceptance of our manuscript. We have made addressed the remaining reviewer concerns to the best of our ability, described in detail below.

Detailed response to reviews:

Reviewer 1 (R1)

1) I could not understand why the authors want to show “a pipeline” instead of successfully developed tools, which are functional promoter cassettes. After publication of the article, do the authors expect for other researchers to follow this scheme to get additional constitutive promoters ???

Our response: We hope the reviewer will agree that we have successfully used our pipeline to identify promoters that function in tobacco and Arabidopsis. The engineering constraints we applied in this manuscript are almost certainly not universally applicable to all potential uses. We expect that others may find that they can make use of our pipeline, likely with their own set of constraints, for identifying promoters with particular patterns of expression (e.g., expressed in a subset of tissues rather than constitutively). In addition, the pipeline can be readily adapted to other organisms, an approach we ourselves have taken in another publication currently under review.

2) Fig. 2 and 3: It is better to show fold change over negative control levels. Presented figures are difficult to guess strength of the signals obtained from positive samples.

Our response: We thank the reviewer for this suggestion and acknowledge that there are other ways to process and represent the data. We maintain that our choice is the clearest for readers, but understand that there is room for disagreement. To facilitate alternative analyses and provide full transparency, all data and scripts (including those for annotating and analyzing the data, as well as for generating figures) are now linked to Zenodo DOI: 10.5281/zenodo.8170303. The data files include individual outputs generated by the plate reader that are consolidated and annotated using R-script before normalized for plotting, and this pre-normalized consolidated dataset is also available for download.

3) Fig. 3A: Signal of RUBY is not easy to see. Pictures need improvements.

Our response: We put a significant amount of effort into photographing the subtle RUBY expression found in our transgenic lines, and, in fact, did a number of ‘blinded’ tests to verify that one plant part was indeed redder than the control. To provide more quantitative measurements of the RUBY signal, we have now added an unbiased analysis of the photographs in the manuscript, where the level of “red” within a region of interest was determined by using the a* axis in the CIELAB color space (Supplementary Figure S4).

4) Fig. 4: Degree of suppression are all around 0.5 fold or more, which, for me, indicates failure of the suppression system. These examples do not look like a practical system.

Our response: We agree that the repression system can be further optimized, and the practical application of the system would depend on the dynamic range desired.

5) I could not find proof of constitutive function of the developed promoters in the figures. This information is necessary.

Our response: For two of the transgenic Arabidopsis lines, we performed qPCR on flower, leaf, root, and seedlings. In all cases, we were able to detect reporter mRNA.

Thank you for again your time and consideration of our work.

Sincerely,

Dr. Jennifer Nemhauser

Department of Biology

University of Washington

jn7@uw.edu

---

## [Reviewer Report]

Dear authors, thanks for having tackled the last remaining issues and delivered such a good quality article. Please apologise for the long time needed. I’m glad to inform you that your manuscript has been accepted for publication.